Development of guidelines for tertiary education institutions to assist them in supporting students with a mental illness: a Delphi consensus study with Australian professionals and consumers

Reavley Nicola J. nreavley@unimelb.edu.au 1 2 3
Ross Anna M. 1 2 3
Killackey Eoin 1 2
Jorm Anthony F. 1 2 3
1 Orygen Youth Health Research Centre, University of Melbourne , Victoria , Australia
2 Centre for Youth Mental Health, University of Melbourne , Victoria , Australia
3 Melbourne School of Population Health, University of Melbourne , Victoria , Australia
Lynskey Michael
Electronic publication date: 2013 Feb 26
Publication date: 2013
Volume: 1
Electronic Location ID: e43
Received 2012 Nov 13; Accepted 2013 Jan 31
Copyright: © 2013 Reavley et al.
Copyright year: 2013
Copyright holder: Reavley et al.
License: This is an open access article distributed under the terms of the Creative Commons Attribution License, which permits unrestricted use, distribution, and reproduction in any medium, provided the original author and source are credited.
License URL: https://creativecommons.org/licenses/by/3.0/

Keywords: Mental health literacy, Tertiary students, Delphi consensus study, Mental illness

Funding: Australian Rotary Health NHMRC Australia Fellowship Funding for the study was provided by Australian Rotary Health and by the NHMRC Australia Fellowship awarded to AFJ. The funders had no role in study design, data collection and analysis, decision to publish, or preparation of the manuscript.

==============================
Background. The age at which most young people are in tertiary education is also the age of peak onset for mental illness. Because mental health problems can have adverse effects on students’ academic performance and welfare, institutions require guidance how they can best provide support. However, the scientific evidence for how best to do this is relatively limited. Therefore a Delphi expert consensus study was carried out with professional and consumer experts.

Methods. A systematic review of websites, books and journal articles was conducted to develop a 172 item survey containing strategies that institutions might use to support students with a mental illness. Two panels of Australian experts (74 professionals and 35 consumers) were recruited and independently rated the items over three rounds, with strategies reaching consensus on importance written into the guidelines.

Results. The overall response rate across three rounds was 83% (80% consumers, 85% professionals). 155 strategies were endorsed as essential or important by at least 80% of panel members. The endorsed strategies provided information on policy, measures to promote support services, service provision, accessibility of support services, relationships between services, other types of support and issues associated with reasonable adjustments. They also provided guidance on the procedures the institutions should have for making staff aware of issues associated with mental illness, mental illness training, support for staff and communicating with a student with a mental illness. They also covered student rights and responsibilities, the procedures the institutions should have for making students aware of issues associated with mental illness, dealing with mental health crises, funding and research and evaluation.

Conclusions. The guidelines provide guidance for tertiary institutions to assist them in supporting students with a mental illness. It is hoped that they may be used to inform policy and practice in tertiary institutions.

Introduction

The 2007 Australian National Survey of Mental Health and Wellbeing estimated that mental illness affects as many as one in four people aged 16 to 24 in any 12-month period (Slade et al., 2009). The age at which most young people are in tertiary education is also the age of peak onset for mental illness, with mental and substance use disorders having their first onset before age 24 in 75% of cases (Kessler et al., 2007; Slade et al., 2009). In Australia, it is estimated that over 50% of young Australians are in tertiary education (Birrell & Edwards, 2007). While many students enjoy and cope well with the transition to higher education, for others it is less easy, possibly due to the competing demands of work, study and family commitments (Andrews & Wilding, 2004). Analysis of data from national surveys reveals that tertiary students are at higher risk of moderate, but not high, psychological distress compared to non-students (Cvetkovski, Reavley & Jorm, 2012). However, having financial problems and working in paid employment increased the risk of distress, an issue that is likely to increase in importance as the participation rate of socio-economically disadvantaged students increases (Cvetkovski, Reavley & Jorm, 2012).

Students with a mental illness face the same challenges as other students as well as unique problems associated with the re-emergence or intensification of symptoms. Mental illness has been shown to affect both exam performance and higher education drop-out rates, with an estimated 86% of individuals who have a psychiatric disorder withdrawing from college prior to completion (Kessler et al., 1995; Andrews & Wilding, 2004; Hysenbegasi, Hass & Rowland, 2005). In an Australian study, Stallman found that Australian students experiencing very high levels of distress were, on average, unable to work or study for eight days within the previous four weeks and had another nine days of reduced capacity for work (2008). Such educational impacts may have lifelong consequences, particularly if students are unable to complete their courses. There is evidence that mental illness in higher education students has increased in recent years, placing pressure on institutions to adequately respond to the needs of the population (Roberts et al., 1999; Kadison, 2004).

Australian tertiary education institutions typically offer a number of services to support students with a mental illness, including counselling services and disability liaison units. However, the great majority of young people with depression and related disorders either do not seek or delay seeking professional help and there is some evidence that they are also reluctant to access formal disability support services in tertiary education institutions (Salzer, Wick & Rogers, 2008; Slade et al., 2009). While depression, anxiety and related disorders are among the leading causes of disability worldwide, the term “disability” is typically seen as relating to physical problems, with the links between mental disorders and disability often poorly understood (World Health Organisation, 2008). Tertiary institution disability services have traditionally been focused on physical disabilities and there is growing acknowledgement of the need for a greater policy and practice focus on support for students with mental illness-related disabilities. This led to the organisation by eight Australian universities of an Australian National Summit on the Mental Health of Tertiary Students in August 2011. The scientific evidence for how best to support students with a mental illness is relatively limited and there is a need to develop and evaluate policies and practices in the area (Salzer, Wick & Rogers, 2008). This is particularly true in the Australian context, as the majority of research has been carried out in Europe and the US, which have somewhat different education systems.

In the context of the limited evidence base, assessing expert consensus offers a way of bringing together available research evidence and best practice in order to enable recommendations and decisions to be made. Such methods have been widely applied in the development of clinical practice guidelines. The most commonly used consensus method is the Delphi process, which has been used to develop mental health first aid guidelines using the expertise of professionals, consumers and carers (Jorm et al., 2008; Kelly et al., 2008; Langlands et al., 2008). These guidelines have been used to revise the content of a Mental Health First Aid training program (Kitchener & Jorm, 2008).

This paper reports on the development of guidelines for tertiary institutions to assist them in supporting students with a mental illness. Once established, the guidelines may be used to inform policy and practice in tertiary institutions.

Methods

The Delphi method

The Delphi process involves a group of experts making private ratings of agreement with a series of statements, feedback to the group of a statistical summary of the ratings, and then another round of rating (Jones & Hunter, 1995). Statements about supporting tertiary students with a mental illness were derived from a search of the lay and scientific literature, and these were presented to a panel of experts in three sequential rounds. Any additional strategies suggested by panel members were included in the subsequent round for all experts to rate. A summary of group ratings was fed back to the panel members after the first two rounds. Panel members could choose to either change or maintain their ratings. In this way, a list of statements that had substantial consensus in ratings was developed, and those statements with low or conflicting ratings discarded.

Panel formation

There were two separate panels, one comprising professionals in the field, include disability liaison unit staff, student counsellors, researchers on student mental health and administrators. The second panel consisted of consumer advocates, i.e. those with current or very recent (within the last 2 years) experience of being a tertiary student with a mental health problem. Professionals were recruited through the Australian and New Zealand Student Services Association (ANZSSA) and through directly approaching student counselling and disability services. ANZSSA has members in 32 of the 39 universities in Australia and 10 of the 60 Technical and Further Education (TAFE) institutes. Participants were limited to Australian institutions due to differences in educational systems. Consumers were recruited by distributing information about the study to consumer organizations associated with mental health issues (including Bluevoices, the consumer forum of beyondblue: the national depression and anxiety initiative). Consumers were offered a voucher worth AUD25 for each round of the survey they completed. The study did not aim to get representative samples of experts, because the Delphi method requires panel members who are information and experience rich rather than representative.

Panel membership numbered 109, with 74 professionals and 35 consumers. 81 panel members were female (66% of the professionals and 91% of the consumers). The median age was 49 years for the professionals and 22 years for the consumers. Of the 74 professionals on the panel, there were 24 disability and support services staff, 23 psychologists, 12 student counsellors, 7 mental health nurses, 7 mental health educators, 7 mental health academic staff, 2 social workers, 1 occupational therapist, and 1 medical practitioner (figures do not add up to 74 due to panel members reporting multiple roles).

Questionnaire development and administration

A systematic literature review was conducted of websites, books and journal articles for strategies about how institutions could support students with mental health problems. This involved a comprehensive search in Google search engines (www.google.com.au, www.google.co.uk, www.google.ca, www.google.com). The following search terms were entered into each: “tertiary education OR higher education OR vocational education OR college OR campus AND depression OR anxiety OR mental disorders OR psychiatric disability”. The first 50 sites for each set of search terms were examined for statements about how institutions could support students with a mental illness. Any links that appeared on these web pages that the authors thought may contain useful information were followed. Relevant journal articles were located on PsycINFO and PubMed, using the search terms “tertiary education” OR university OR college AND student* AND “mental disorder” [Title/Abstract] OR “mental illness” [Title/Abstract] OR “psychiatric disability” AND policy OR procedures OR services.

We obtained suggestions for how institutions could support students with a mental illness from 59 websites (50 tertiary education institution sites), 3 pamphlets and 21 journal articles. Suggestions were limited to “actionable items” – actions that could be taken by tertiary institutions. Just over 80% of initially identified sources provided at least one suggestion for the initial questionnaire. The majority of strategies came from tertiary institution websites. In addition, the questionnaire content was informed by a small number of strategies suggested by the working group to fill perceived gaps in the questionnaire’s content. One of the authors (AR) carried out the literature review, analysed the information gathered from these sources and wrote the suggestions up as individual survey items. This document was presented to a working group comprising the authors, who screened the items to ensure they fitted the definition of actions that tertiary institutions could take to support students with a mental illness, were comprehensible, and had a consistent format, while remaining as faithful as possible to the original wording of the information. After several draft surveys, the group produced a list of 172 items that formed the first survey sent to panel members.

The Round 1 survey was organized into 9 sections (see Table 1). Panel members were asked to rate the importance of each item. The rating scale used was: essential, important, depends, unimportant, should not be included, don’t know. The Round 1 survey also included comment boxes that allowed panel members to give feedback after each section. To analyse the comments that panel members had written in the first round questionnaire, one of the authors (AR) read through all the comments and wrote them up as draft strategies. The working group evaluated the suggested draft strategies to determine whether they were original ideas that had not been included in the first round questionnaire. Any strategy that was judged by the group to be an original idea was included as a new item to be rated in the second round questionnaire. Panel members completed the questionnaires online using SurveyMonkey. The study was approved by the Human Research Ethics Committee of the University of Melbourne (Ethics ID: 1034964).

Table 1 Round 1 survey sections and number of items.

Section	Number of items	
Policy	22	
Policy content	11	
Policy development and implementation	8	
Communicating the policy	3	
Support services	26	
Awareness of support services	4	
What support services should provide	8	
Accessibility of support services	8	
Relationships with other services	6	
Other types of support	10	
Reasonable adjustments	7	
Staff	56	
Mental illness awareness	20	
Mental illness training	14	
Support for staff	3	
Communicating with students with a mental illness	14	
Support from teaching staff	3	
Students	29	
Student rights and responsibilities	11	
Mental illness awareness	18	
Dealing with mental health crises	7	
Funding	5	
Research and evaluation	12	

Statistical analysis

On completion of each round, the survey responses were analysed by obtaining percentages for the professional and consumer panels for each item. The following cut-off points were used:

Criteria for accepting an item

• If at least 80% of both the professional and consumer panels rated an item as essential or important as a guideline for institutions supporting students with a mental illness, it was included in the guidelines.

Criteria for re-rating an item

Panel members to rerate an item in the next round if:

• 80% or more of the panel members in one group rated an item as essential or important

• 70–79% of panel members in both groups rated an item as either essential or important

Criteria for rejecting an item

Any items that did not meet the above conditions were excluded.

Results

The response rate of those who took part in all three rounds was 83% (80% professionals, 85% consumers). See Table 2 for the number of panel members who completed each round.

Table 2 Participant numbers for each round of the survey.

	Round 1
n	Round 2
N (%)	Round 3
N (%)	
Consumer	35	30 (86)	28 (80)	
Professional	74	63 (85)	63 (85)	
Total	109	93 (85)	91 (83)	

See Fig. 1 for an overview of the numbers of items that were included, excluded, created and re-rated in each round of the survey. Across three rounds, 155 strategies were rated as essential or important by at least 80% of the both panels (see Supplemental Table 1). Overall, ratings of whether items were essential or important were similar across the consumer and professional panels, with a correlation of r = .75.

Figure 1 Overview of items included, excluded, created and re-rated in each round of the survey.

One of the authors (AR) prepared a draft of the guidelines by grouping items of similar content under specific headings. The guidelines retained the original wording of the items as much as possible, whilst remaining easy to read. The draft guidelines were then given to panel members for final comment, feedback and endorsement. No changes were asked for by panel members at this stage.

Table 3 Key points for tertiary education institutions to facilitate improved educational outcomes for students with a mental health problem.

Have a policy around supporting students with a mental health problem	
• The institution should have a mental health policy covering mental health promotion, mental illness prevention and services for students with a mental illness.

• The mental health policy and its implementation should be driven by senior management in partnership with students with mental illnesses, staff from different areas of the institution, student associations and representatives of outside services.

• The institution should have a strategy for communicating its mental health policy to staff and students.

	
Provide support to students with a mental health problem	
• The disability office should make all staff aware of the range of services they provide to assist and educate staff supporting students with a mental illness.

• Support services should develop a mental health promotion strategy which covers prevention, early identification, stigma reduction, availability and access to services.

• Support services should provide all staff and students with education on mental illness.

• The institution’s support services should adopt an easy access and “no wrong door” policy to entry for assessment and treatment of mental health problems.

	
Provide reasonable adjustment for students with a mental illness	
• Staff should be provided with information about making reasonable adjustments for assessments.

• The process for getting reasonable adjustments should be as simple as possible and advice should be available to students if needed.

	
Have procedures for making staff and students aware of issues around mental illness	
• These should include signs and symptoms, causes and treatments, the importance of prevention and early intervention and how to support students with a mental illness in ways that promote recovery.

• Support services staff should receive appropriate and ongoing professional development and training in relation to mental illnesses.

• The institution should provide staff with training and information about the following:

○ The use of non-judgemental listening skills when talking with students about their personal problems.

○ How to respond when a student discloses a mental illness to them, including which things are supportive and which are unhelpful.

○ Techniques for promoting motivation and self-esteem in students with mental illnesses.

○ Curriculum design, development and delivery strategies that facilitate inclusive and effective learning for students with mental illnesses.

○ Classroom, examination and assignment adjustments that can be made for a student with a mental illness.

• Make students aware of their rights and responsibilities

• Staff should be informed about how to handle mental health crisis situations.

	
Interact with students with a mental illness in a manner that maintains respect, dignity, confidentiality and equity	
• When a student discloses that they have personal issues such as a mental illness, confidentiality should be respected unless there is an immediate danger to the person or to others in withholding that information.

• If the student has a mental illness, staff should not make assumptions, but rather ask the student what support, if any, they might need.

• Staff should explore any challenges or barriers to successful learning with students with a mental illness.

	
Allocate resources to funding and evaluation	
• Adequate funds should be allocated to provide support services to students with a mental illness.

• Institutions should seek funding opportunities that can be used to help develop and enhance support services for students with a mental illness.

• The institution’s mental health services should be subject to ongoing research and evaluation of their service provision.

	

The final guidelines (see Table 3 for a summary and Supplemental Figure 1 for the full guidelines) provide information and advice on how tertiary institutions should support students with a mental illness. They cover policy, measures to promote support services, service provision, accessibility of support services, relationships between services, other types of support and issues associated with reasonable adjustments. They provide guidance on the procedures the institutions should have for making staff aware of issues associated with mental illness, mental illness training, support for staff and communicating with a student with a mental illness. They also cover student rights and responsibilities, the procedures the institutions should have for making students aware of issues associated with mental illness, dealing with mental health crises, funding and research and evaluation.

Discussion

The project aimed to identify strategies that could be used by tertiary institutions to support students with a mental illness. Overall, 155 strategies were endorsed from a comprehensive range of suggestions. The endorsed strategies were written into a guidelines document which is freely available to tertiary institutions in order to inform policy and practice.

With 90% of the original items rated as essential or important by both panels, there was a high level of agreement between consumers and professionals on the importance of the items. One notable area of difference related to the statement that support service hours should include regular evening and weekend hours. This was endorsed by 73% of consumers but only 40% of professionals. Other areas of difference also related to services provided to students, including the recommendation that the organisation’s support services should be one organisational unit; that the institution should offer short courses for students with a mental illness on how best to manage their illness while fulfilling the student role; that students with a mental illness should be provided with individualised support for their education goals; that students with a mental illness should have access to vocational support; that students should have the opportunity to join an ongoing peer support group; and that funding should be available for students who have had a period of hospitalisation due to mental illness to catch up with their studies. All these items received notably higher endorsement ratings from consumers than from professionals. It is likely that these differences reflect health professionals’ views (many of whom were student counsellors or disability service officers) on what is practical in their organisations given available resources. It may also reflect the views expressed in a number of comments that such services should be available for all students, not just those with a mental illness. Several respondents commented that they thought students with a mental illness would not want to be singled out in this way.

The issue of what is practical in an institution may also explain differences in endorsement ratings of items relating to research and evaluation of services. More consumers than staff endorsed items relating to research into the prevalence of mental health problems in students; research into the needs of those who are typically more reluctant to access services; research into new models of service delivery; and the establishment of a working group to raise, discuss and advocate on issues affecting students with a mental illness. There were several comments about the applicability of general community research to those in tertiary institutions and for the need for inter-institutional collaboration in this area.

Items that received notably lower endorsement by consumers than professionals included those recommending that campus security procedures should exist to deal with students in a mental health crisis; that the institutions support service should have a way of prioritising appointments for urgent situations; and that there should be collaboration (with a student’s permission) with external mental health agencies. The recommendations that all staff receive mental health training and that teaching staff should let their students know they can be approached if the students have problems also received notably lower endorsements from consumers than professionals. This may reflect a limited belief in the helpfulness of lecturers or teachers as sources of help for mental health problems, something that has been seen in general community surveys (Reavley & Jorm, 2011).

While the guidelines provide guidance for tertiary institutions on how to support students with a mental illness, how they are implemented is critical to their usefulness. This will obviously vary according to institution. In August 2011, an Australian National Summit on the Mental Health of Tertiary Students, organised by representatives from eight universities was held in Melbourne. Summit attendees included student counsellors, disability service personnel, researchers and senior management from the majority of Australian universities and some Technical and Further Education institutions. Implementation of the guidelines was discussed by working groups and a number of ideas emerged, including the use of the guidelines as a way to evaluate current practice, with a focus on accountability and evaluation of success. There was a suggestion that they could be incorporated into the process of planning and prioritising activities in the area of mental health and wellbeing and some groups recommended their inclusion into an institution-wide approach. The guidelines were welcomed as a source of new ideas, evidence and opportunities for benchmarking. Future research should investigate the impact of the guidelines on policies and practice in tertiary institutions and may also be used as a basis for designing interventions aimed at improving mental health in tertiary students.

A number of groups felt that the guidelines could be used as a tool to engage senior management and help them understand the importance of mental health issues. Given the necessity for funding to support the implementation of some of the recommendations outlined in the guidelines, senior management support is critical. It was also felt that they would be useful in getting buy-in from other staff members, including academic staff. The recruitment of “champions” to consider how the guidelines may interface with existing policies was a further recommendation.

Strengths of the Delphi approach include the ability to include individual panellists across diverse professional and geographical locations without the need for face-to-face meetings. Moreover, the anonymous nature of the Delphi process ensures that no single expert can dominate the consensus process, a factor that may be particularly important when including both professionals and young consumers. However disadvantages of the approach include an inability of panellists to meet and discuss uncertainties or ambiguities in, for example, the construction or wording of the questionnaires used. The success or otherwise of the Delphi process depends on the panel of experts chosen to participate and lack of diversity in panel members may limit the usefulness of the findings. In this study, male consumers, who may have different attitudes to help seeking for mental health problems to female consumers (Reavley, McCann & Jorm, 2012) were represented in low numbers.

Conclusions

In conclusion, the guidelines provide guidance for tertiary education institutions on how to facilitate improved educational outcomes for students with a mental illness. It is hoped that they be used to inform policy and practice in these institutions. Further research should assess the implementation of the guidelines in a variety of institutions, including universities and TAFEs.

Supplemental Information

Supplemental Table 1 Supplemental item 1: Items that received 80% consensus across the consumer, carer and clinician panels

Click here for additional data file.

Supplemental Figure 1 Supplemental item 2: Guidelines for tertiary education institutions to facilitate improved educational outcomes for students with a mental illness

Click here for additional data file.

We would like to thank ANZSSA and beyondblue for their assistance in recruiting panel members. We would also like to acknowledge the organisers of the National Summit on the Mental Health of Tertiary Students, in particular, Jonathan Norton and Matthew Brett.

Additional Information and Declarations

Competing Interests

Author Contributions

Ethics

Anthony Jorm is an Academic Editor for PeerJ.

Nicola J. Reavley conceived and designed the experiments, performed the experiments, analyzed the data, wrote the paper.

Anna M. Ross performed the experiments, analyzed the data, wrote the paper.

Eoin Killackey conceived and designed the experiments, comments on manuscript.

Anthony F. Jorm conceived and designed the experiments, performed the experiments, analyzed the data, comments on manuscript.

The following information was supplied relating to ethical approvals (i.e. approving body and any reference numbers):

University of Melbourne Human Research Ethics Committee (Ethics ID: 1034964).

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
