# Peer review of "Development of guidelines for tertiary education institutions to assist them in supporting students with a mental illness: a Delphi consensus study with Australian professionals and consumers"

_PeerJ, doi:10.7717/peerj.43_

## Round 0.1 · original submission · Minor Revisions

Thank you for this submission. I would like to apologize for the delay in returning reviews and a decision to you. Despite this delay, I do hope you find the attached reviews helpful.

In my view, this very well written paper addresses an important - yet relatively under-researched topic. Given the relative lack of research in this area, the application of Delphi consensus methods is both appropriate and informative. In addition to the comments made by reviewers, I did wonder if the authors might consider it useful to provide a (very brief) description of what they see as the 'nest steps', both for research and for practice, in this area.

·

Basic reporting

The article is really clearly written. It includes a useful and appropriate introduction both to the problem (lack of guidelines for supporting students in tertiary education) and the solution - using the DELPHI consensus method. Structure is clear and follows usual practice. The article definitely meets PeerJ standards.

Experimental design

Research question is clearly defined. The methods are mostly explained in enough detail. My only comment about improvement is I would like to know a bit more about the data extraction process, the authors refer to this as a systematic literature review, but give no detail about what information was extracted, how many websites/documents that were initially identified were not used, and how many members of the team chose which data to include in the initial survey that was then seen by the working party.

Validity of the findings

The tables and figures are very useful in giving a clear view of the DELPHI process. This is also made clear in the summary of items included. There is a useful section in the discussion outlining work that has already been done in how the guidelines might be implemented. One omission that I feel needs to be addressed is the lack of a strengths/limitations section about the process itself. One shortcoming that should definitely be discussed is the very few male consumers who took part - there is evidence that young men may have different attitudes to help seeking and may develop different coping strategies, therefore it seems vital to me that more men would ideally have been involved in the development of guidelines. Did the authors do anything to try and get more young male consumers involved?

Additional comments

This was a very enjoyable paper to read, not least because of its immediate and direct application to practice.

·

Basic reporting

Basic reporting requirements appear to have been met.

Experimental design

The approach is appropriate and good response rates have been obtained. I have some reservations about the extent to which the focus of the paper, consensus guideline development for supporting students with mental illness, is within the scope of the journal (biological and medical sciences). Ethical requirements have been met. The research question is relevant and meaningful but gets lost in the Introduction. While generally sharp and focussed, I could not understand the point of the section on the purpose of disability services and the fact that they concentrate on physical complaints. Often these type of services are provided within the one setting. Why is this argument important to the focus of the paper anyway? It is very difficult for the reader to discern, the source and approach for the consumer respondents, other than that they were collected from consumer groups.

Validity of the findings

The validity of the findings and the conclusions get lost somewhat. There is an inordinate amount of discussion about where the two groups disagreed and the final results are only presented as a supplementary file. While the title promises the presentation of guidelines, the findings are presented more as thematic areas. Thus, it is difficult to ascertain the general format and final content of the guidelines at a glance.

Additional comments

I would suggest a general sharpening of the paper and a summary of the final guidelines presented as a table so that they can be accessed at a quick glance. Hopefully this will enhance consideration by practitioners.

---

## Round 0.2 · accepted · Accept

Thank you for your revision and for addressing the (relatively minor) comments and suggestions from reviewers. These revisions have adequately addressed reviewer concerns and I am happy to accept the manuscript in its current form.